# Left Atrial Appendage Closure in Atrial Fibrillation Patients with Cancer

**DOI:** 10.3390/jcm13216514

**Published:** 2024-10-30

**Authors:** David Zweiker, Jutta Bergler-Klein, Lukas Fiedler, Gabor G. Toth, Reinhard Achleitner, Alexandra Schratter, Guenter Stix, Harald Gabriel, Ronald K. Binder, Martin Rammer, Michael Pfeffer, Paul Vock, Brigitte Lileg, Clemens Steinwender, Kurt Sihorsch, Florian Hintringer, Agne Adukauskaite, Martin Martinek, Thomas Sturmberger, Klemens Ablasser, Andreas Zirlik, Daniel Scherr

**Affiliations:** 1Division of Cardiology, Medical University of Graz, 8036 Graz, Austria; gabor.g.toth@medunigraz.at (G.G.T.); klemens.ablasser@medunigraz.at (K.A.); andreas.zirlik@medunigraz.at (A.Z.); daniel.scherr@medunigraz.at (D.S.); 2Third Department for Cardiology and Intensive Care, Clinic Ottakring, 1160 Vienna, Austria; 3Department of Cardiology, University Clinic of Internal Medicine II, Medical University of Vienna, 1090 Vienna, Austria; jutta.bergler-klein@meduniwien.ac.at (J.B.-K.); guenter.stix@meduniwien.ac.at (G.S.); harald.gabriel@meduniwien.ac.at (H.G.); 4Department of Internal Medicine, Cardiology and Nephrology, Hospital Wiener Neustadt, 2700 Wiener Neustadt, Austria; lukas.fiedler@wienerneustadt.lknoe.at (L.F.); michael.pfeffer@wienerneustadt.lknoe.at (M.P.); 5Department of Cardiology, Clinic Floridsdorf, 1210 Vienna, Austria; reinhard.achleitner@gesundheitsverbund.at (R.A.); alexandra.schratter@gesundheitsverbund.at (A.S.); 6Department of Internal Medicine II, Klinikum Wels-Grieskirchen, 4600 Wels, Austria; ronald.binder@klinikum-wegr.at (R.K.B.); martin.rammer@klinikum-wegr.at (M.R.); 7Department of Internal Medicine III, University Hospital St. Pölten, 3100 St. Pölten, Austria; paul.vock@aon.at (P.V.); office@lileg-kardiologie.at (B.L.); 8Department of Cardiology, Kepler University Hospital, 4020 Linz, Austria; hcsteinwender@hotmail.com (C.S.); kurt.sihorsch@kepleruniklinikum.at (K.S.); 9Department of Internal Medicine III, Medical University of Innsbruck, 6020 Innsbruck, Austria; florian.hintringer@tirol-kliniken.at (F.H.); agne.adukauskaite@tirol-kliniken.at (A.A.); 10Department of Internal Medicine II, Elisabethinen Hospital, 4020 Linz, Austria; martin.martinek@ordensklinikum.at (M.M.); thomas.sturmberger@ordensklinikum.at (T.S.); 11Department of Cardiology, Cardiovascular Research Institute Maastricht (CARIM), Maastricht University Medical Centre, 6229 Maastricht, The Netherlands

**Keywords:** left atrial appendage closure, anticoagulation, atrial fibrillation, complications, cancer, malignancy, cardio-oncology

## Abstract

**Background**: There are limited data about left atrial appendage closure (LAAC) in patients with cancer. We therefore sought to compare the outcome after LAAC in patients with vs. without cancer in a multicentre registry. **Methods**: In this sub-analysis of the prospective Austrian LAAC Registry, we analysed consecutive patients undergoing LAAC to assess the relationship between baseline characteristics and outcome in patients with vs. without cancer. Inverse probability weighting was performed to adjust for differences in baseline characteristics. **Results**: A total of 486 consecutive patients from 9 centres with a median age of 75 years (IQR 70–79 years; 35.8% female) were included. Fifty-seven patients (11.7%) had a history of cancer. The median CHA_2_DS_2_-VASc and HAS-BLED scores were similar in both groups (median [IQR], 4 [4–6] vs. 5 [3–5], *p* = 0.415; 4 [3–4] vs. 3 [3–4], *p* = 0.428 in cancer vs. other patients). Cancer patients were significantly older, and anaemia and gastrointestinal bleeding were significantly more common. Major procedural complications occurred in 5.3% vs. 7.0% (*p* = 0.276) of patients. The cumulative five-year survival rates were 80.7% and 84.8% in cancer vs. other patients (adjusted hazard ratio for death 1.29 [95% CI 0.67–2.48], *p* = 0.443). There were also no differences in one-year survival (96.1% vs. 94.0%, *p* = 0.582) and five-year event-free survival (64.9% vs. 74.4%, *p* = 0.124). **Conclusions**: In daily clinical practice, LAAC has already been accepted as a treatment option in patients with cancer. This retrospective analysis shows that short-term and adjusted long-term complications are similar in patients with vs. without cancer undergoing LAAC.

## 1. Introduction

Atrial fibrillation (AF) is the most common relevant arrhythmia in developed countries [1]. AF facilitates thrombus formation and cardioembolic stroke, mainly due to stasis and prothrombotic endovascular transformation of the left atrium. Oral anticoagulation (OAC) is recommended in most patients with comorbidities, based on the CHA_2_DS_2_-VA score [1]. OAC, however, increases the risk of bleeding and may therefore not be suitable in patients with major or recurrent bleeding events, vessel malformation, or other contraindications. Furthermore, a proportion of patients develop ischaemic stroke, despite adequate OAC intake [2]. As 90% of intracardiac thrombi form in the left atrial appendage (LAA) [3], interventional and surgical methods for LAA closure (LAAC) as an alternative to OAC have been developed [4]. According to current European Society of Cardiology (ESC) guidelines for the treatment of AF, interventional LAAC may be considered in patients with AF and contraindications for long-term anticoagulant treatment to prevent ischaemic stroke and thromboembolism (indication class IIb, level of evidence C) [1].

Patients with malignancies show a high prevalence of AF: atrial arrhythmias are four times more prevalent in cancer patients compared with the general population [5], and new-onset AF is associated with cancer incidence [6]. The management of such patients may be complicated by a high risk of cardioembolic events and simultaneously increased risk of bleeding events [7]. This specific problem was addressed in the first cardio-oncology guidelines of the ESC in 2022, recommending an individual approach regarding antithrombotic management [7]. In cancer patients with high embolic risk and prohibitive bleeding risk, LAAC is considered a suitable alternative to OAC, if life expectancy is considered > 12 months (recommendation class IIb, level of evidence C). As cancer survival is tremendously improving due to novel oncology therapies, cardiovascular complications are increasingly important competitive risks influencing long-term outcome [8]. Worse survival has been observed in cancer patients with AF, and furthermore, the CHA_2_DS_2_-VASc score may underestimate the risk of thromboembolism, as well as high bleeding risk on the other side [7,9].

There are currently limited data about patients with malignancy and AF receiving LAAC. We performed a sub-analysis of the Austrian national prospective multicentre LAAC registry and sought to investigate the efficacy and safety of LAAC in AF patients with cancer compared with those without cancer.

## 2. Materials and Methods

This is a sub-analysis of the prospective Austrian LAAC Registry (NCT03409159) [10,11], which includes all patients who underwent LAAC in Austria. The main purpose of this analysis was the comparison of baseline characteristics, indication, and outcome in patients with vs. without cancer. The registry was approved by the institutional review board of the Medical University of Graz (29-355 ex 16/17).

### 2.1. Recruitment and Procedure

Patients with AF, high thromboembolic risk, and either contraindication, intolerance, or ineffectiveness of OAC were evaluated for LAAC at each of the referral centres in Austria. Attending physicians at the respective centres were free to decide the suitability of LAAC according to the institutes’ protocols and international guidelines [1,12]. According to the decision of the ethics committee, written informed consent was not necessary due to the retrospective nature of the registry. The primarily used devices were Watchman™, Watchman FLX™ (Boston Scientific, Marlborough, MA, USA), Amplatzer Cardiac Plug™, Amplatzer Amulet™ (Abbott Laboratories, North Chicago, IL, USA), and LAmbre™ (Lifetech Scientific, Shenzhen, China). Post-procedural management was tailored according to international guidelines and the patients’ individual risk profile. In accordance with international recommendations [1], intensified antithrombotic therapy was performed for 1–6 months after LAAC, followed by transoesophageal echocardiography and a de-escalation of antithrombotic therapy in most patients. Follow-up was scheduled at each centre’s discretion.

### 2.2. Inclusion and Exclusion Criteria

The only inclusion criterion of this analysis is a history of LAAC in Austria. We defined no exclusion criteria.

### 2.3. Data Collection

The Austrian national LAAC registry includes parameters recommended by the latest European Heart Rhythm Association (EHRA) and European Association of Percutaneous Cardiovascular Interventions (EAPCI) position paper [12] and experimental variables. Data entry was performed by either a local representative or an external reviewer. All centres were visited at least once by an external reviewer with monitoring of a random sample to ensure adequate data quality. Mortality data from the Austrian government’s population registry were included. Patient inclusion was complete in all centres until the end of 2021 and in 3 of 9 centres until mid-2023.

### 2.4. Endpoints

We defined major periprocedural complications as all complications before hospital discharge requiring invasive intervention. Access site complications were documented if invasive intervention was necessary. We defined ischaemic stroke according to guidelines [13]. For long-term outcome assessment, one-year and five-year rehospitalisation was analysed. Furthermore, we evaluated the 5-year rate of echocardiography-confirmed device thrombus formation, stroke, major bleeding (Bleeding Academic Research Consortium types 3–5) [14] or LAAC-required hospitalisation.

### 2.5. Statistical Analysis

For the primary analysis, we compared patients with vs. without cancer regarding their short-term and long-term complication rates. All patients with a history of malignancy, regardless of the status at the time of LAAC (active or in remission), were summarized in the “cancer” group. The “other” group included all remaining patients. We expressed parameters as count (proportion), mean ± standard deviation, or median (interquartile range), as appropriate. Depending on the presence of normal distribution, calculated by the Shapiro–Wilk test, either Student’s T test or the Wilcoxon signed-rank test were used for univariable analysis. Long-term events were adjusted for differences in baseline characteristics by inverse probability weighing. Cumulative survival or event rates were compared with the log-rank test. A two-sided *p* value of <0.05 was considered significant. For statistical analysis, R 4.3.1 (The R Project, Vienna, Austria) and RStudio 2023.12.1 + 402 “Ocean Storm” (Posit Software, PBC, Boston, MA, USA) were used.

## 3. Results

A total of 486 patients undergoing LAAC between November 2010 and January 2024 at 9 Austrian centres were included in this analysis. Of those, 11.7% (*n* = 57) had a history of cancer.

### 3.1. Baseline Characteristics

The median age was 75 years and 35.8% were female (Table 1). The median (IQR) body mass index was 27 (24–30) kg/m^2^ and body surface area was 1.92 ± 0.20 m^2^. The median (IQR) CHA_2_DS_2_-VASc score was 5 (3–5), CHA_2_DS_2_-VA score was 4 (3–5) and HAS-BLED score was 3 (3–4).

Patients with cancer were significantly older than the other patients and had similar thromboembolic and bleeding risk, as estimated in the CHA_2_DS_2_-VASc, CHA_2_DS_2_-VA, and HAS-BLED scores. Anaemia was significantly more prevalent in cancer patients. Gastrointestinal bleeding as an indication for LAAC was more frequent in cancer patients and occurred most often in the lower gastrointestinal tract. In contrast, haemorrhagic stroke was significantly less common in cancer patients. A higher NT-proBNP level was observed in cancer patients. Otherwise, there were no differences in baseline laboratory parameters between both groups (Table 1).

In patients with cancer, the tumour was considered active in 14.0% of patients (n = 8) while the remaining patients were in remission at the time of LAAC (Table 2). One-fifth of patients (21.1%, n = 12) suffered from complications affecting daily life prior to LAAC. Metastatic cancer was only prevalent in 3.5% of cases. Half of the patients had cancer originating from the urinary system (50.9%), mainly the prostate. A quarter had a gastrointestinal neoplasia, and the remaining patients had breast, skin, haematologic, thyroid, or lung cancer. Cancer therapy incorporated surgery in most cases (83.7%), radiotherapy (68.3%), chemotherapy (16.7%), or hormone therapy (4.8%).

Bleeding was the most common indication that finally led to the decision to perform LAAC, with no differences between groups (66.7% vs. 68.3%). Thromboembolism despite OAC had occurred in 9.7% of cases, and other reasons were similar in both groups. The decision to perform LAAC was based on cancer-related complications in 14.0% of cancer patients (n = 8) due to bleeding from the gastrointestinal tract (10.5%) or from the urinary tract (3.5%).

### 3.2. Procedural Details and Outcome

There were no differences in preprocedural planning and device selection between cancer and other patients (Table 1). In most cases, Amplatzer and Watchman devices were used. LAAC was successfully performed in 94.7% of cancer patients and in 97.0% of other patients without differences between groups (*p* = 0.418). The procedure lasted for about 1 h in both groups, and the fluoroscopy time and contrast medium were similar.

Procedural complications occurred in 5.3% of cancer patients and in 7.0% of other patients (*p* = 0.785, Table 3). There were no significant differences in specific types of procedural complications between both groups. Major procedural complications that led to permanent disability were rare (less than 2% in both groups), and <1% of patients died.

Post-interventional antithrombotic therapy was tailored according to the patients’ individual thromboembolic and bleeding risk, and was similar in both groups. Short-term antithrombotic treatment included dual antiplatelet therapy (55.8%), OAC (13.2%), and single antiplatelet therapy (8.8%). Due to the high bleeding risk, 4.7% received no antithrombotic therapy at all after discharge. After 2–6 months, most patients received single antiplatelet therapy (47.3%) or no therapy (36.2%). NOAC therapy was maintained for more than 6 months in 6.8% of patients.

The median (IQR) follow-up duration was 30 (13–53) months. Bleeding, stroke or LAAC-related hospitalisations were similar between groups, with a five-year cumulative event rate of 35.1% in cancer patients and 25.6% in other patients (*p* = 0.114). The cumulative bleeding rate was numerically higher in cancer patients (27.8% vs. 13.4%), but this difference was not statistically significant. The cumulative proportion of patients that had experienced a stroke at 5 years was similar in both groups (1.8% in cancer patients vs. 3.9% in no-cancer patients). LAAC-associated hospitalisations occurred only in seven non-cancer patients (1.6%) and one cancer patient (1.8%). Other patients had admissions due to device thrombus (n = 3), Dressler syndrome after implantation (n = 1), heart surgery for the capture of mobilized devices (n = 1), device patency associated with cardioembolic stroke (n = 1), and surgical revision of a groin wound after femoral puncture (n = 1). One cancer patient had gastrointestinal bleeding associated with post-procedural dual antiplatelet therapy.

One-year survival was 96.1% in cancer patients and 94.0% in other patients (*p* = 0.580). Five-year cumulative survival (80.7% vs. 84.8%, *p* = 0.410, Figure 1A) and five-year event-free survival (64.9% vs. 74.6%, *p* = 0.110, Figure 1B) were also similar between both groups. Five-year survival in patients with active cancer (n = 8) was 62.5% compared with 84.7% in the remaining patients (*p* = 0.092). In the inverse-probability weighted analysis, neither one-year survival, five-year survival, nor five-year event-free survival were significantly different, without a tendency favouring any of the groups (central illustration).

## 4. Discussion

This study shows that (1) a relevant proportion of patients undergoing LAAC had a history of cancer (13.3%), (2) baseline characteristics in cancer vs. other patients differed regarding age, gastrointestinal bleeding, anaemia, and haemorrhagic stroke, and (3) baseline-adjusted outcomes were similar between groups.

As both AF and cancer are associated with higher age and specific comorbidities, they are often concomitant conditions [5]. There is even a higher risk of developing cancer immediately after new-onset AF [6] and vice versa [15]. Cancer may significantly affect further prognosis and may require multiple surgical and medical interventions [5]. Furthermore, both thromboembolism and bleeding may be more frequent in cancer patients. All these factors complicate the decision to start OAC. Therefore, the current ESC cardio-oncology guidelines suggest an individual approach based on specific bleeding and thrombotic risk, taking into account the possibility of LAAC in selected patients [7]. This analysis shows that, in daily clinical practice, LAAC was often perceived as an alternative to OAC in patients with cancer.

Cancer patients were significantly older compared with the other patients, highlighting the age-dependent cancer risk [16]. This association has also been demonstrated in most previous analyses of cancer patients undergoing LAAC [17,18,19].

Gastrointestinal bleeding, either directly from the tumour or as side effect of its treatment, is a significant problem in cancer patients. In this analysis, especially a history of gastrointestinal bleeding was significantly more common in cancer patients. Radiation therapy-associated proctitis with lower gastrointestinal bleeding led to the decision to perform LAAC in many cases. On the other hand, intracerebral bleeding was less common in cancer patients. Notably, no cerebral metastases were documented in any of the included patients. Rates of anaemia and bleeding were also higher in other studies [18,19]. Unfortunately, this study was not powered to detect differences in different cancer types, as a high incidence of AF and worse survival have been reported in breast cancer patients [20].

Procedural characteristics and short-term outcomes were similar between cancer and non-cancer groups. Consequently, it is reasonable to interpret the procedural risk in cancer patients similarly to other LAAC patients.

While cancer may generally limit the prognosis of the general population, there was no sign towards reduced long-term survival in the cancer population in this study. Without adjustment for other baseline characteristics, the cumulative bleeding rate at 5 years was doubled in patients with cancer (27.8% vs. 13.4%), leading to a 10% higher absolute risk of total long-term complications (35.1% vs. 25.6%). However, this difference did not persist after adjustment for baseline characteristics. We argue that treating centres selected patients with suitable prognosis, as defined by guidelines [7], to undergo LAAC. In the current literature, there is conflicting evidence regarding the influence of a cancer history on long-term outcome after LAAC [17,18,21], but patient-level analyses found no differences in outcome when adjusting for detailed baseline data [17,21].

To the authors’ knowledge, this is the first analysis regarding cancer patients from a complete multicentric national LAAC registry, while previous studies reported on one-centric [21] or two-centric [17] experiences or retrospective analyses from the Nationwide Readmissions Database (NRD) of the United States [18,19]. Due to the inclusion of three different device manufacturers and low-volume centres, data from this analysis have high external objective validity. Additionally, the analysis was carefully adjusted for baseline characteristics, which flattened differences in the long-term outcomes between groups.

### 4.1. Gaps in Literature

While some observational data of patients with cancer undergoing LAAC have been published, there is currently no study available focusing on patients receiving LAAC due to complications directly attributed to their malignancy, such as bleeding from the tumour or side-effects from radiotherapy. In those patients, oral antithrombotic therapy often cannot be maintained because of recurrent bleeding events, putting them at high risk of thromboembolic events. LAAC may be especially helpful in those patients. This study population was underrepresented in this analysis (eight patients) and deserves further investigation in the future.

Furthermore, the experience with LAAC and active cancer must further be evaluated to identify specific active malignancy-related complications. Patients with cerebral metastases have a higher risk of complications. It is currently unclear if LAAC is beneficial in these patients, taking into account competing risks, such as the patients’ overall prognosis. In this analysis, no patients with cerebral metastases were included.

Finally, a randomized trial of patients with cancer undergoing LAAC or medical antithrombotic therapy is urgently needed to understand the effect of LAAC in this special patient population.

### 4.2. Limitations

Despite its advantages of a nation-wide registry with consecutive patient inclusion, this analysis may present limitations in the selection of cancer patients undergoing LAAC with all forms of associated bias, especially in the estimation of cancer prognosis and risks leading to the selection of patients. As this analysis was performed post hoc, information bias may have led to the misclassification of group allocation, undocumented short- and long-term complications, and therefore skewed outcome interpretation. Especially inactive cancer diagnosed and treated many years before LAAC may have been overlooked. There was variability in performing follow-up visits between centres, which may have further influenced outcome. While complete mortality data have been ascertained, missing data may underestimate the reported rate of non-fatal events.

## 5. Conclusions

A substantial proportion of patients undergoing LAAC had a history of cancer. Although these patients differed significantly from the other patients regarding their baseline characteristics, short-term complications and importantly long-term survival were similar in both groups. The findings of this retrospective analysis support the use of LAAC in selected cancer patients with a high bleeding risk to optimize their prognosis and overall survival.

## Figures and Tables

**Figure 1 jcm-13-06514-f001:**
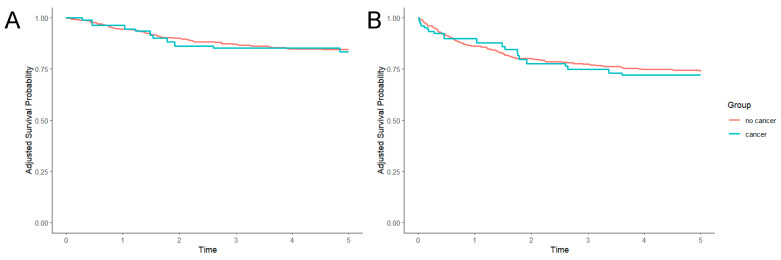
Non-adjusted five-year survival curves for patients with vs. without cancer. (**A**): five-year survival; (**B**): five-year survival free from long-term adverse events (embolism, stroke, bleeding, LAAC-related hospitalisation).

**Table 1 jcm-13-06514-t001:** Baseline characteristics.

Parameter	Total Population (n = 486)	Patients with Cancer (n = 57)	Other Patients (n = 429)	*p* Value
**Baseline demographics**				
female gender	35.8% (n = 174)	33.3% (n = 19)	36.1% (n = 155)	0.769
age (years)	75 (70–79)	78 (74–81)	75 (70–79)	**<0.001**
body mass index (kg/m^2^)	27 (24–30)	28 (25–31)	27 (24–30)	0.120
body surface area (m^2^)	1.92 ± 0.20	1.94 ± 0.22	1.92 ± 0.20	0.490
**Comorbidities**				
CHA_2_DS_2_-VASc score	5 (3–5)	4 (4–6)	5 (3–5)	0.415
CHA_2_DS_2_-VA score	4 (3–5)	4 (3–5)	4 (3–5)	0.248
HAS-BLED score	3 (3–4)	4 (3–4)	3 (3–4)	0.428
arterial hypertension	88.9% (n = 432)	93.0% (n = 53)	88.3% (n = 379)	0.374
diabetes mellitus	28.4% (n = 138)	22.8% (n = 13)	29.1% (n = 125)	0.352
congestive heart failure	26.5% (n = 129)	31.6% (n = 18)	25.9% (n = 111)	0.424
stroke or TIA	42.4% (n = 206)	38.6% (n = 22)	42.9% (n = 184)	0.570
ischemic stroke	25.9% (n = 126)	19.3% (n = 11)	26.8% (n = 115)	0.203
haemorrhagic stroke	25.1% (n = 122)	12.3% (n = 7)	26.8% (n = 115)	**0.021**
intracerebral bleeding	20.8% (n = 101)	14.0% (n = 8)	21.7% (n = 93)	0.225
subarachnoid bleeding	3.7% (n = 18)	0% (n = 0)	4.2% (n = 18)	0.248
subdural bleeding	4.9% (n = 24)	1.8% (n = 1)	5.4% (n = 23)	0.341
epidural bleeding	1.0% (n = 5)	1.8% (n = 1)	0.9% (n = 4)	0.460
gastrointestinal bleeding	38.3% (n = 186)	52.6% (n = 30)	36.4% (n = 156)	**0.020**
lower gastrointestinal tract	27.4% (n = 133)	42.1% (n = 24)	25.4% (n = 109)	**0.011**
upper gastrointestinal tract	22.8% (n = 111)	31.6% (n = 18)	21.7% (n = 93)	0.129
chronic kidney disease	22.0% (n = 107)	26.3% (n = 15)	21.4% (n = 92)	0.398
dialysis	1.9% (n = 9)	0% (n = 0)	2.1% (n = 9)	0.608
chronic liver disease	5.4% (n = 22)	10.7% (n = 3)	5% (n = 19)	0.188
anaemia	41.8% (n = 282)	59.6% (n = 34)	39.4% (n = 169)	**0.009**
prior blood transfusion	34.6% (n = 168)	43.9% (n = 25)	33.3% (n = 143)	0.098
coronary artery disease	40.7% (n = 198)	47.4% (n = 27)	39.9% (n = 171)	0.316
cerebral artery disease	12.8% (n = 62)	12.3% (n = 7)	12.8% (n = 55)	1.000
periphery artery disease	10.5% (n = 51)	12.3% (n = 7)	10.3% (n = 44)	0.645
chronic obstructive pulmonary disease	12.6% (n = 61)	14.0% (n = 8)	12.4% (n = 53)	0.673
paroxysmal AF	34.0% (n = 165)	22.8% (n = 13)	35.4% (n = 152)	0.073
vascular malformation				0.091
lower gastrointestinal tract	5.6% (n = 27)	8.8% (n = 5)	5.1% (n = 22)	
upper gastrointestinal tract	4.7% (n = 23)	3.5% (n = 2)	4.9% (n = 21)	
cerebral	4.7% (n = 23)	3.5% (n = 2)	4.9% (n = 21)	
other	0.4% (n = 2)	1.8% (n = 1)	0.2% (n = 1)	
prior acute coronary syndrome	11.9% (n = 58)	14.0% (n = 8)	11.7% (n = 50)	0.663
prior pulmonary embolism	2.1% (n = 10)	3.5% (n = 2)	1.9% (n = 8)	0.331
prior peripheral embolism	1.9% (n = 9)	1.8% (n = 1)	1.9% (n = 8)	1.000
**echocardiography**				
LVEF				0.592
normal	74.1% (n = 361)	70.2% (n = 40)	74.6% (n = 321)	
35–50%	19.3% (n = 94)	21.1% (n = 12)	19.1% (n = 82)	
<35%	6.4% (n = 31)	8.8% (n = 5)	6.1% (n = 26)	
severe aortic stenosis	1.0% (n = 5)	3.5% (n = 2)	0.7% (n = 3)	0.107
severe mitral regurgitation	6.0% (n = 29)	10.5% (n = 6)	5.4% (n = 23)	0.134
severe tricuspid regurgitation	5.6% (n = 27)	3.5% (n = 2)	5.8% (n = 25)	0.757
**laboratory**				
erythrocytes (T/L)	4.26 ± 0.69	4.15 ± 0.72	4.28 ± 0.68	0.230
haemoglobin (g/dL)	12.5 (10.9–14.0)	12.5 (10.8–13.6)	12.7 (10.9–14.1)	0.436
haematocrit (%)	37 (33–42)	37 (34–41)	38 (33–42)	0.910
platelets (G/L)	217 (173–261)	211 (167–234)	218 (174–264)	0.122
NT-ProBNP (ng/L)	907 (404–2104)	1421 (748–3610)	829 (361–2041)	**0.022**
creatinine (mg/dL)	1.10 (0.90–1.41)	1.10 (0.95–1.61)	1.10 (0.90–1.40)	0.440
eGFR (ml/min/1.73 m^2^)	74.8 (67.8–81.4)	73.1 (62.9–80.8)	75.0 (68.0–81.5)	0.216
ASAT (U/L)	24 (20–30)	26 (22–30)	24 (19–30)	0.279
ALAT (U/L)	20 (15–28)	18 (15–25)	20 (15–28)	0.383
INR	1.1 (1.0–1.2)	1.1 (1.0–1.2)	1.1 (1.0–1.2)	0.708
aPTT (sec)	35 (31–40)	34 (30–41)	35 (31–40)	0.488
albumine (g/L)	41 (36–44)	44 (43–45)	41 (36–44)	0.103
total protein (g/L)	71 (66–74)	72 (69–75)	70 (65–74)	0.304
total cholesterol (mg/dL)	152 (122–185)	158 (121–181)	151 (122–186)	0.655
LDL (mg/dL)	86 (62–110)	91 (67–107)	85 (62–113)	0.905
triglycerides (mg/dL)	98 (73–141)	98 (72–128)	98 (73–143)	0.443
**Indication for LAAC**				
indication group				0.669
bleeding	68.1% (n = 331)	66.7% (n = 38)	68.3% (n = 293)	
other	22.0% (n = 108)	26.3% (n = 15)	21.4% (n = 92)	
thromboembolism	9.7% (n = 47)	7% (n = 4)	10% (n = 43))	
primary indication for LAAC				0.214
gastrointestinal bleeding	33.5% (n = 163)	43.9% (n = 25)	32.2% (n = 138)	
intracranial Bleeding	26.3% (n = 128)	15.8% (n = 9)	27.7% (n = 119)	
bleeding under OAC	6.6% (n = 32)	7% (n = 4)	6.5% (n = 28)	
stroke	6.2% (n = 30)	3.5% (n = 2)	6.5% (n = 28)	
other	4.5% (n = 22)	1.8% (n = 1)	4.9% (n = 21)	
anaemia	3.7% (n = 18)	8.8% (n = 5)	3% (n = 13)	
predisposition for bleeding	3.7% (n = 18)	7% (n = 4)	3.3% (n = 14)	
other contraindication for OAC	3.5% (n = 17)	5.3% (n = 3)	3.3% (n = 14)	
embolism despite OAC	3.5% (n = 17)	3.5% (n = 2)	3.5% (n = 15)	
OAC intolerance	2.9% (n = 14)	0% (n = 0)	3.3% (n = 14)	
patient preference	2.1% (n = 10)	1.8% (n = 1)	2.1% (n = 9)	
epistaxis	1.6% (n = 8)	0% (n = 0)	1.9% (n = 8)	
requirement for triple therapy	1.6% (n = 8)	1.8% (n = 1)	1.6% (n = 7)	
dialysis	0.2% (n = 1)	0% (n = 0)	0.2% (n = 1)	
contraindication for OAC	18.7% (n = 91)	19.3% (n = 11)	18.6% (n = 80)	0.858
cancer-related LAAC indication	1.6% (n = 8)	14.0% (n = 8)	N/A	N/A
device				0.742
Amplatzer	57.8% (n = 281)	61.4% (n = 35)	57.3% (n = 246)	
Watchman	41.6% (n = 202)	38.6% (n = 22)	42% (n = 180)	
LAmbre	0.4% (n = 2)	0% (n = 0)	0.5% (n = 2)	
isolated procedure	90.9% (n = 442)	87.7% (n = 50)	91.4% (n = 392)	0.099

eGFR: estimated glomerular filtration rate; LAAC: left atrial appendage closure; LDL: low-density lipoprotein; LVEF: left ventricular ejection fraction; OAC: oral anticoagulation. Significant *p* values (<0.05) are written bold.

**Table 2 jcm-13-06514-t002:** Details of cancer patients.

Parameter	Patients with Cancer
active cancer	14.0% (n = 8)
in remission	86.0% (n = 49)
complications affecting daily clinical life *	21.1% (n = 12)
spread	
localised	96.5% (n = 55)
metastatic	3.5% (n = 2)
metastatic cancer	3.5% (n = 2)
**Origin**	
urologic/gynaecologic	50.9%
prostate	28.1%
kidney	8.8%
bladder	5.3%
uterus	5.3%
seminoma	1.8%
vulva	1.8%
gastrointestinal	24.6%
colorectal	19.3%
gastroduodenal	3.5%
tongue	1.8%
breast	14.0%
skin	10.5%
basalioma	7.0%
melanoma	3.5%
haematologic	5.3%
B cell lymphoma	3.5%
myelodysplastic syndrome	1.8%
thyroid	3.5%
lung	1.8%
**Cancer therapy**	
surgery	83.7%
radiotherapy	68.3%
active radiotherapy at time of LAAC	2.4%
chemotherapy	16.7%
active chemotherapy at time of LAAC	4.8%
hormone therapy	4.8%
**Indication for LAAC**	
cancer-related indication	14.0%
bleeding due to radiotherapy induced proctitis	8.7%
haematuria due to kidney or bladder cancer	3.5%
bleeding due to gastrointestinal stromal tumour	1.8%

* complications affecting daily clinical life were defined as complications from the tumour disease leading to irreversible disability and/or frequent rehospitalizations at time of LAAC.

**Table 3 jcm-13-06514-t003:** Procedural and long-term outcome in patients with vs. without cancer.

Parameter	Patients with Cancer	Other Patients	*p* Value
fluoroscopy time (min)	15 (12–20)	15 (10–22)	0.970
dose area product (µGym^2^)	4795 (1857–8651)	3868 (1325–8443)	0.208
contrast medium (mL)	95 (61–120)	97(65–135)	0.598
procedure duration (min)	63 (51–86)	61 (50–88)	0.851
successful deployment of LAAC device	94.7% (n = 54)	97.0% (n = 416)	0.418
prolonged post-procedural stay (>2 nights)	35.1% (n = 20)	26.6% (n = 114)	0.276
**Procedural complications**	5.3% (n = 3)	7.0% (n = 30)	0.785
pericardial tamponade	1.8% (n = 1)	1.9% (n = 8)	1.000
access site complications	0% (n = 0)	2.9% (n = 12)	0.565
requiring surgery	0% (n = 0)	1.9% (n = 8)	
requiring thrombin injection	0% (n = 0)	0.7% (n = 3)	
requiring angioseal	0% (n = 0)	0.2% (n = 1)	
stroke	1.8% (n = 1)	1.2% (n = 5)	1.000
death	0% (n = 0)	0.5% (n = 2)	1.000
heart surgery	0% (n = 0)	0.5% (n = 2)	1.000
interventional retrieval of dislocated LAAC device	0% (n = 0)	0.5% (n = 2)	1.000
cardiopulmonary resuscitation	0% (n = 0)	0.5% (n = 2)	1.000
device embolization	0% (n = 0)	0.5% (n = 2)	1.000
**Long-term follow-up**			
cumulative one-year survival	96.1%	94.0%	0.582
cumulative five-year survival	80.7%	84.8%	0.412
cumulative 5-year event or death rate	35.1%	25.6%	0.124
device thrombus formation	5.3%	6.1%	
stroke	0.0%	2.8%	
bleeding	17.5%	9.1%	
LAAC associated hospitalisation	0.0%	1.2%	

## Data Availability

The data presented in this study are available on request from the corresponding author.

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
