# Peer review of "Left Atrial Appendage Closure in Atrial Fibrillation Patients with Cancer"

_jcm, 2024, doi:10.3390/jcm13216514_

Round 1
Reviewer 1 Report
Comments and Suggestions for Authors
1. The paper did not answer the study question: compare outcome after LAAC in patients with vs without cancer.
2. Lack of detail on recruitment: How were patients identified, included and excluded?
3. Follow-up was "scheduled at each centre's discretion" => introduce variability in monitoring, poetically influence outcomes.
4. Missing details on data verification: is there any data validation steps to ensure the accuracy of information recorded from multiple centers?
5. last of data regarding certain types of cancers, stages of cancers, which could have a significant impact on the interpretation of the findings.
6. Imbalance comparison, only 14% patients with active cancer vs 86% in remission.
7. Data were duplicated. Unclear about labs collected (For example: NT-ProBNP, ..)
8. Last of deal on gaps in literature
9. There are some redundancies in the text as well as the data table.
10. Lack of details on specific types of complications in cancer patients.
Comments on the Quality of English Language1. The paper did not answer the study question: compare outcome after LAAC in patients with vs without cancer.
2. Lack of detail on recruitment: How were patients identified, included and excluded?
3. Follow-up was "scheduled at each centre's discretion" => introduce variability in monitoring, poetically influence outcomes.
4. Missing details on data verification: is there any data validation steps to ensure the accuracy of information recorded from multiple centers?
5. last of data regarding certain types of cancers, stages of cancers, which could have a significant impact on the interpretation of the findings.
6. Imbalance comparison, only 14% patients with active cancer vs 86% in remission.
7. Data were duplicated. Unclear about labs collected (For example: NT-ProBNP, ..)
8. Last of deal on gaps in literature
9. There are some redundancies in the text as well as the data table.
10. Lack of details on specific types of complications in cancer patients.
Author Response
- The paper did not answer the study question: compare outcome after LAAC in patients with vs without cancer.
Answer: Thank you for this interesting comment. Indeed, we performed an extensive analysis to compare the outcome after LAAC in patients with vs. without cancer. We present this statement in the abstract in the “Background” section (“We therefore sought to compare the outcome after LAAC in patients with vs. without cancer in a multicentre registry”), the “Methods” section (“… we analysed consecutive patients undergoing LAAC to assess the relation between baseline characteristics and outcome in patients with vs. without cancer”), the “Results” section (“Major procedural complications occurred in […]. Cumulative five-year survival was […] in cancer vs. remaining patients […]. There were also no differences in one-year survival […] and five-year event-free survival […].”), and the “Conclusions” section (“Short-term and adjusted long-term complications are similar in patients with vs. without cancer undergoing LAAC”). There are also comparisons between patients with vs. without cancer throughout the manuscript, especially in the “Conclusions” section (“[…] short-term complications and importantly the long-term survival were similar in both groups.”) [lines 307-313].
For clarification, we added the following information into the “Materials and Methods” section of the manuscript: “The main purpose of this analysis was the comparison of baseline characteristics, indication and outcome in patients with vs. without cancer.” [lines 92-94]; “For the primary analysis, we compared patients with vs. without cancer regarding their short-term and long-term complication rate. […] All patients with a history of malignancy, regardless of the status at time of LAAC (active or in remission), summarized in the “cancer” group. The no-cancer group includes all remaining patients.” [lines 133-136].
- Lack of detail on recruitment: How were patients identified, included and excluded?
Answer: This is a very important question, as the inclusion and exclusion criteria must be taken into account when applying the results of the study to other populations.
The only inclusion criterion is the performance of LAAC in Austria. This information is found in the Abstract (“In this sub-analysis of the prospective Austrian LAAC Registry, we analysed CONSECUTIVE patients undergoing LAAC […]”) and in the Methods section: “This is a sub-analysis of the prospective Austrian LAAC Registry, which includes all patients that underwent LAAC in Austria” [lines 91-92]. Details about the indication of LAAC are found in section 2.1. (“Patients with AF, high thromboembolic risk and either contraindication, intolerance or ineffectiveness of OAC were evaluated for LAAC at each of the referral centres in Austria. Attending physicians at the respective centres were free to decide the suitability of LAAC according to the institutes’ protocol and international guidelines.” [lines 97-100]).
We now added a new section 2.2. “Inclusion and exclusion criteria”, where we state our only inclusion criterium (“The only inclusion criterion of this analysis is a history of LAAC in Austria. We defined no exclusion criteria.” [lines 111-113].
- Follow-up was "scheduled at each centre's discretion" => introduce variability in monitoring, poetically influence outcomes.
Answer: Variability in follow-up throughout the centres is indeed a major problem that introduces bias in the data. Unfortunately, due to the nature of this analysis (which is pan-national), we cannot ensure regular follow-up in all centres.
We tried overcoming this limitation by focussing on data that was available in all patients: short-term outcome (until discharge) and mortality (which was assessed from the national survival database).
We added this limitation into the Limitations section: “There was variability in performing follow-up visits between centres, which may have influenced outcome.” [lines 304-305]
- Missing details on data verification: is there any data validation steps to ensure the accuracy of information recorded from multiple centers?
Answer: To harmonize the data entry and reduce variations between centres in the assessment of outcome parameters, we chose to perform all data input into the registry in most centres by one reviewer (DZ). He performed the complete data entry in 7 of 9 centres. In the remaining two centres, the data input was performed partly by a local representative, but also validated by the same reviewer by random sampling. Data quality was further enhanced by extracting mortality data from the government’s survival database.
We added the following information into the section 2.3. Data collection: “Data entry was performed by either a local representative or an external reviewer. All centres were visited at least once by an external reviewer with random sampling to ensure adequate data quality.” [lines 118-120]
- last of data regarding certain types of cancers, stages of cancers, which could have a significant impact on the interpretation of the findings.
Answer: Thank you for this comment. Patients with many different types of cancer were included in this analysis, and the absolute number of patients with a specific cancer type was low. Furthermore, the exact stage was not available in some patients, for example if they were in remission for many years before LAAC. While it may be interesting to link the individual cancer types or stages with exact complications, we chose not to enclose this information in the manuscript due to ethical reasons.
- Imbalance comparison, only 14% patients with active cancer vs 86% in remission.
Answer: Thank you for this correct observation. As this is an analysis from real-world patients undergoing LAAC, there is no possibility for us to balance the absolute number of comorbidities or cancer stages.
- Data were duplicated. Unclear about labs collected (For example: NT-ProBNP, ..)
Answer: Thank you for this comment, Table 2 was accidentally duplicated. We now removed the duplicated rows [line 168].
- Last of deal on gaps in literature
Answer: This is a very good point. We added a paragraph about gaps in literature into the Discussion section [new section 4.1., lines 283-299]
- There are some redundancies in the text as well as the data table.
Answer: We thank the reviewer for this comment. We removed some data that is mentioned in both the text and the data table [lines 158-177].
- Lack of details on specific types of complications in cancer patients.
Answer: We thank the reviewer for this important comment. We performed further analyses and expanded the paragraph about specific types of long-term complications after LAAC in cancer patients, taking also into account stroke events [lines 204-217].
Reviewer 2 Report
Comments and Suggestions for Authors
The manuscript presents a sub-analysis of a prospective multicenter national registry about LAA closure procedures for atrial fibrillation patients with cancer. The authors compared the baseline characteristics, procedural complications and outcomes between patients with vs. without cancer. No significant differences in complications and outcomes were found between the two groups.
The study is appropriately designed, well-written, with an adequate sample size (486 patients). It is the first multicenter registry for this specific population. The study provides clear, interesting and important data regarding LAA closure procedures in patients with cancer, which are frequently seen in daily clinical practice.
Table 2 needs to be corrected (double-written).
Author Response
- The manuscript presents a sub-analysis of a prospective multicenter national registry about LAA closure procedures for atrial fibrillation patients with cancer. The authors compared the baseline characteristics, procedural complications and outcomes between patients with vs. without cancer. No significant differences in complications and outcomes were found between the two groups.
The study is appropriately designed, well-written, with an adequate sample size (486 patients). It is the first multicenter registry for this specific population. The study provides clear, interesting and important data regarding LAA closure procedures in patients with cancer, which are frequently seen in daily clinical practice.
Table 2 needs to be corrected (double-written).
Answer: We thank the reviewer for the comments. We corrected Table 2 accordingly.
Reviewer 3 Report
Comments and Suggestions for Authors
I have a few questions and comments about this manuscript.
1. The table was probably accidentally duplicated 2, since the data is repeated in it.
2. The text after the table often repeats the data that is displayed in the table with the same numbers. What's the point of that? This only increases the volume of the manuscript. Everything is perfectly clear from the table. Limit yourself to extracting the essence from this data. For example, lines 133-142; 146-159.
3. The central illustration should be removed from the body of the manuscript.
4. Why do the authors consider long-term results as a combined endpoint? It is likely that if each event is considered separately, the bleeding rate will be higher in the group of patients with cancer. This seems to me to be an important point.
Author Response
I have a few questions and comments about this manuscript.
- The table was probably accidentally duplicated 2, since the data is repeated in it.
Answer: Thank you for this comment, we adapted Table 2 accordingly.
- The text after the table often repeats the data that is displayed in the table with the same numbers. What's the point of that? This only increases the volume of the manuscript. Everything is perfectly clear from the table. Limit yourself to extracting the essence from this data. For example, lines 133-142; 146-159.
Answer: We thank the reviewer for this comment and removed data from these sections that are already written in the table [lines 158-177].
- The central illustration should be removed from the body of the manuscript.
Answer: We do not know how the editorial team handles the central illustration in the manuscript. In order for the reviewers to see it, we still left the central illustration in the manuscript, but we moved it to the top.
- Why do the authors consider long-term results as a combined endpoint? It is likely that if each event is considered separately, the bleeding rate will be higher in the group of patients with cancer. This seems to me to be an important point.
Answer: We thank the reviewer for this very important question. This is an analysis of a multicentric registry. Post-procedural management and follow up procedures vary substantially between centres, which introduces bias into the data. However, we can be sure about the mortality data as it is associated with the national survival database. Therefore, most of the reported long-term outcome variables only include survival. As we cannot exclude underreporting or missing data at follow-up completely, we tried to minimize our assumptions from non-fatal events at follow up and chose to use a combined endpoint only.
Due to the reviewer’s suggestion, we performed an additional analysis and looked at the cumulative event rate. We found a numerically doubled bleeding rate in patients with cancer before adjustment for baseline characteristics. We added this information, as well as information about the stroke rate (which was tendentially lower in cancer patients), to the Results section [lines 206-209] and also to the Discussion [lines 265-275].
Round 2
Reviewer 3 Report
Comments and Suggestions for Authors
The authors have made edits to the manuscript, in accordance with the comments. There are no more comments.
Author Response
The authors have made edits to the manuscript, in accordance with the comments. There are no more comments.
- Answer: We thank the reviewer for this comment.